# Reactive Comb Polymer Compatibilized Immiscible PVDF/PLLA Blends: Effects of the Main Chain Structure of Compatibilizer

**DOI:** 10.3390/polym12030526

**Published:** 2020-03-02

**Authors:** Xin Yang, Jinxing Song, Hengti Wang, Qingqing Lin, Xianhua Jin, Xin Yang, Yongjin Li

**Affiliations:** 1College of Materials, Chemistry and Chemical Engineering, Hangzhou Normal University, No. 2318 Yuhangtang Rd., Hangzhou 311121, China; yangxin@stu.hznu.edu.cn (X.Y.); hengti-wang@hznu.edu.cn (H.W.); lqq0770709@163.com (Q.L.); 2Transfar Zhilian Co. Ltd., Hangzhou 311215, China; songjinxing@126.com (J.S.); gfjxh@163.com (X.J.); 11002@etransfar.com (X.Y.)

**Keywords:** comb structure, main chain, compatibilizing effect, co-continuous

## Abstract

The compatibilizer with double comb structure has a superior compatibilizing effect for immiscible polymer blends due to the symmetrical structure on both sides of main chains. Extensive study related to the architectural effects of compatibilizer on the compatibilization has mainly focused on the side chains. We investigated the influence of the compatibilizer-main-chain structure on the compatibilizing effect for immiscible poly(vinylidene fluoride)/poly(L-lactic acid) (PVDF/PLLA) blends. Two reactive-comb compatibilizers with polystyrene (PS) and polymethylmethacrylate (PMMA) as main chains and PMMA as the side chains have been synthesized. PS is immiscible with both PLLA and PVDF, while PMMA is miscible with PVDF. It was found that both compatibilizers can improve the compatibility between the PLLA and PVDF, with different compatibilization effects. In the PVDF/PLLA (50/50) blends, 1 wt.% poly(styrene-co-glycidyl methacrylate)-graft-poly(methyl methacrylate) (RC–SG) tuned the morphology from the droplet-in-matrix structure to the co-continuous structure, while the blends with poly(methyl methacrylate-co-glycidyl methacrylate)-graft-poly(methyl methacrylate) (RC–MMG) kept the sea-island structure with even 3 wt.% loading. Moreover, RC–SG induces a wider co-continuous interval range than RC–MMG. The co-continuous structure obtained by RC–SG was also more stable than that by RC–MMG. It was further found that RC–SG-compatibilized PVDF/PLLA blends exhibit higher mechanical properties than the RC–MMG-compatibilized blends.

## 1. Introduction

Polymer blending has become an important way to prepare new high-performance polymeric materials over the past 40 years [1,2,3,4,5]. However, most commercial polymer blends are immiscible because of their high molecular weights and unfavorable interactions, and, thus, form multiphase structures [6,7,8]. Consequently, a variety of compatibilizers have been developed and applied for immiscible polymer blends [9,10,11,12]. A great quantity of research has indicated that the reactive-comb-like compatibilizers usually exhibit better compatibilizing effects due to the double-comb structures. The double-grafted side chains of the compatibilizers can stabilize the dynamic balance of the neighboring phases at the interface [13,14,15,16]. This technique is based on the in-situ formation of double-grafted copolymers at polymer–polymer interface, which substantially lowers interface tension, suppresses particle coalescence, and ultimately exhibits enhanced physical properties [17,18,19,20].

Considering the compatibilization mechanism of the reactive-comb polymers for a specific blend, the side chains (grafts A and grafts B) of the comb compatibilizers should exhibit high affinity (or specific interactions) with the component of the polymer blends, and, therefore high- compatibilization efficiency can be achieved, as shown in Scheme 1. It is obvious that the side chains take the critical role for the final compatibilization by the respective entanglements of each graft with the molecular chains of the component while the main chain simply bonds to the side chains to form one molecular structure [21,22,23,24,25,26]. Therefore, investigations have been focused on the length of the side chain and the graft density of the comb compatibilizers effects on the compatibilizations [27,28,29,30,31]. Attention has seldom been paid to the main chain effects on the compatibilization effects. Wang et al. have synthesized a comb-like copolymer with polystyrene (PS) as the main chain, i.e., poly(styrene-*co*-glycidyl methacrylate)-*graft*-poly(methyl methacrylate) (P((St-*co*-GMA)-*g*-MMA)), and applied it for compatibilizing PVDF/PLLA immiscible blends [32,33]. The epoxide groups of this compatibilizer can react to the terminal carboxyl groups of PLLA during the melt-blending, forming in situ a double-grafted copolymer (i.e., (PLLA-*g*-PS-*g*-PMMA)). Interestingly, they found that the PS main chain of the double-grafted copolymer would collapse to form nanomicelles at the PVDF–PLLA interface. The nanomicelles could further self-assemble to create Janus nanomicelles (JNMs) with one side rich in PMMA and the other in PLLA side chains, and thus enhancing the compatibilization. This interfacial JNMs compatibilization was attributed to the unique quality of PS main chain, indicating that the main chain of compatibilizer also plays an indispensable role in compatibilization.

To clarify the role of main chain of compatibilizer in the compatibilization, we designed two reactive-comb copolymers with almost identical structures, except the main chain, namely poly(styrene-*co*-glycidyl methacrylate)-graft-poly(methyl methacrylate) (RC–SG) and poly(methyl methacrylate-*co*-glycidyl methacrylate)-graft-poly(methyl methacrylate) (RC–MMG) by the “grafting through” method. The comparison has been made for the immiscible poly(vinylidene fluoride)/poly(L-lactic acid) (PVDF/PLLA) blends. It was found that both RC–SG and RC–MMG exhibit effective compatibilization effects, but RC–SG easily induces co-continuous structures of the blends and a wider co-continuous interval range than RC–MMG.

## 2. Experimental Section

### 2.1. Materials

The PVDF used in this study was purchased from Kureha Chemicals (Tokyo, Japan). And PLLA was supplied from Nature works (Blair, NE, USA). The reactive copolymers RC–SG or RC–MMG were synthesized by copolymerizing MMA macromer, glycidyl methacrylate (GMA), and St or MMA. The specific synthesis routes of RC–SG and RC–MMG are shown in Appendix A. The ^1^H-NMR and IR spectra of RC–MMG is shown in Appendix A. In the ^1^H-NMR spectrum, peaks a, c, d, g, h and i represent MMA characteristic signals; peaks b, e, f are GMA characteristic signals. In Appendix A, the MMA characteristic peaks (1731 cm^−1^, 1270 cm^−1^, 1243 cm^−1^, 1191 cm^−1^, 1144 cm^−1^) and epoxy characteristic peak (908 cm^−1^) appear in the infrared spectrum. Appendix A shows the ^1^H-NMR and IR spectra of RC–SG. In the ^1^H-NMR spectrum, peaks a, b and g represent characteristic signals on St; peaks i, f and h represent the characteristic signals of epoxy group; peaks c, d and e are characteristic signals of MMA macromer. Each peak can also be found in the IR spectrum (Appendix A), the characteristic absorption peaks near 3000 cm^−1^ are stretching vibration of –CH on St; peak 908 cm^−1^ is epoxy characteristic signal; 1730 cm^−1^, 1270 cm^−1^, 1244 cm^−1^, 1190 cm^−1^ and 1146 cm^−1^ are MMA-characteristic absorption bands. Therefore, we confirmed the synthesis of reactive-comb polymers (RC–SG and RC–MMG) with the same structure, except for the main chain.

The nomenclature of the copolymers is as follows: the side chains of the two reactive copolymers are PMMA (*M_n_* = 4000) and the reactive compatibilizers contain 20 wt.% of the GMA monomer. Detailed characteristics of the copolymers used are listed in Table 1.

Gel permeation chromatography (GPC) (Wyatt T-rEX, Santa Barbara, CA, USA) was carried out to determine the number-average molecular weights (*M_n_*) and dispersities (Đ) by using two MZ-Gel SD plus 10.0 μm bead-size columns (10^3^ and 10^5^ Å) and Optilab T-rEX detector. THF was used as the mobile phase at a flow rate of 1 mL/min at 35 °C. The system was calibrated with narrow molecular weight distribution polystyrene standards from 2000 to 10^6^ g/mol. The elution diagrams were analyzed using the ASTRA 6 software from Wyatt Technology.

### 2.2. Sample Preparation

All the materials were dried in a vacuum oven at 80 °C for at least 24 h before mixing. The PVDF/PLLA (50/50, w/w) blends without and with various RC–SG or RC–MMG contents were produced by melt-blending directly into the batch mixer (Haake Polylab QC), (Thermo Fisher Scientific, Waltham, MA, USA) with a rotation speed of 50 rpm at 190 °C for 10 min. Subsequently, the resulted blends were compression-molded at 200 °C under 10 MPa for 5 min and cooled to 35 °C by circulating water.

### 2.3. Characterization

#### 2.3.1. Scanning Electron Microscopy (SEM)

SEM observations were performed on a field-emission SEM machine (Hitachi S-4800, Tokyo, Japan) operated at a voltage of 5 kV. Before the observation, the entire specimens were cryo-fractured in liquid nitrogen. Then, the specimens were coated with gold by plasma deposition.

#### 2.3.2. Transmission Electron Microscopy (TEM)

To detect the structure and distribution of RC–SG or RC–MMG, reactive-blended samples were observed by a transmission-electron microscopy system (Hitachi HT-7700, Tokyo, Japan) operated at an accelerating voltage of 100 kV. All samples were stained by RuO_4_ for 4 h to enhance the contrast before the TEM observation.

#### 2.3.3. Mechanical Properties

Charpy impact tests were performed according to GB/T 16420-1996 standard on an impact tester (Sartec SS-3700, Taiwan, China). The samples are molded by an injection-molding machine with the size of 80.0 × 10.0 × 4.0 mm^3^. The tensile testing was on an Instron universal-material testing-system (Instron, Norwood, MA, USA) at room temperature, and the tension rate was 10 mm/min. The specimens were punched out into dumbbell shapes, with the size of 18 mm, 3 mm, 0.5 mm (length, width, thickness). All tensile and impact results are shown in Appendix A.

#### 2.3.4. Dynamic Mechanical Analysis (DMA)

DMA was carried out using DMA Q-800 (TA Instrument, New Castle, PA, USA) under a nitrogen atmosphere. Firstly, the sample was cut up into a size of 8 mm, 6.3 mm, 0.5 mm (length, width, thickness). Then, all specimens were tested at a frequency of 5 Hz, an amplitude of 4 µm, and a heating rate of 10 °C /min from −50 to 200 °C.

## 3. Results

### 3.1. Morphologies

To investigate the influence of RC–SG and RC–MMG as compatibilizers on the morphology of immiscible PVDF/PLLA blends, the fracture surfaces of samples were analyzed by SEM, as shown in Figure 1. For the pure PVDF/PLLA (50/50) blend, PVDF forms domains that disperse in the PLLA matrix (Figure 1a) due to the much higher melt viscosity and density of PVDF than of PLLA. The PVDF/PLLA system is a classic thermodynamic immiscible blend. The PVDF domain size varies from 5 to 50 µm, indicating a high-interfacial tension between the phases. Besides, some clear gaps were observed at the PVDF–PLLA interface for their weak interfacial adhesion. When 0.5 wt.% RC–SG was added, the PVDF domains became more refined, with number-average diameters about 2.5 µm, indicating the effective compatibilization effects of RC–SG. Continuingly increasing the dosage of RC–SG, the morphology of the mixture has changed from a sea-island structure to a co-continuous structure at 1 wt.% loading. Similarly, the size of PVDF droplet decreases with the increasing RC–MMG content, and the interfacial adhesion between the phases becomes stronger. This means that RC–MMG is also a good compatibilizer for the PVDF/PLLA blends. However, as opposed to the blends compatibilized by RC–SG, the PLLA/PVDF blends keep the typical sea-island structure, even with 3 wt.% loading. The domain size decreases continuously with increasing RC–MMG loadings. Therefore, it is concluded that both RC–SG and RC–MMG compatibilize the PVDF/PLLA blends effectively, while the compatibilization modes differ. It is obvious that the epoxide groups on the main chain can react with the end-carboxylic groups of PLLA during the melt-blending, and the double grafts form to compatibilize the blends. The side chain and epoxy content of RC–SG and RC–MMG are equal, except for the trunk chain. It is indicated that the significant main-chain functions during compatibilizing the immiscible PVDF/PLLA system with almost the same side chains. 

Figure 2 shows the fractured surfaces of PVDF/PLLA blends at different components with and without 1 wt.% compatibilizer. For PVDF/PLLA blends without compatibilizers, only the PVDF/PLLA (65/35) blend is a co-continuous structure and the phase size is about 30 µm (Figure 2b). It must be noted that the figure magnification of this blend is not equal to other compatibilized blends due to its large co-continuous phase size. The co-continuous interval does not change at all with the addition of 1 wt.% RC–MMG. However, for PVDF/PLLA blends compatibilized with 1 wt.% RC–SG, a wider co-continuous interval from PVDF/PLLA = 50/50 to 75/25 was identified, as shown in Figure 3. The significant expansion of the co-continuous interval can be clearly observed for the RC–SG-compatibilized blends, as compared with the blends without compatibilizers or compatibilized by RC–MMG.

The morphology of highly immiscible polymer blends is not stable due to the phase coalescence in the melt state [34,35]. Such coalescence is originated from the high-interfacial tension between the phases. Therefore, the morphological stability of polymer blends in melt indicates the effects of the compatibilizers, and this can usually be confirmed by the annealing experiments [36,37]. The thermal stabilities of the co-continuous morphology of the PVDF/PLLA (65/35) blends with 1% RC–SG and RC–MMG as compatibilizers are compared in Figure 4. All samples were annealed under vacuum at 200 °C for 1 h under static conditions. For PVDF/PLLA/RC–MMG (65/35/1) blend, the phase morphology has changed greatly from the original co-continuous structure to the sea-island structure, indicating the poor interfacial stability (Figure 4a,a_1_). In contrast, there was little change to the structure of the PVDF/PLLA (65/35) blend compatibilized by RC–SG after annealing, as evidenced in Figure 4b,b_1_. This means that the blend with RC–SG has better interfacial stability and the co-continuous structure is stable upon the annealing. Favis et al. have illustrated the morphology evolution of the co-continuous blends during annealing [38]. The driving force of coarsening during annealing is the capillary pressure from the minor phase domains, which is proportional to the interfacial tension. Due to the lower interfacial tension of the blend with RC–SG than that with RC–MMG, it exhibits a superior thermal stability. All the morphological analysis indicates that the RC–SG induces the co-continuous structure and the formed morphology is more stable than RC–MMG.

### 3.2. Mechanical Properties

Figure 5 reveals the mechanical properties of the representative samples compatibilized by RC–SG and RC–MMG. Markedly enhanced ductility is observed for the PVDF/PLLA blend containing RC–SG and RC–MMG (Figure 5a). The elongation at break of the PVDF/PLLA (50/50) blend without any compatibilizer is 3.6%, compared with 159.0% for the blend with 0.5 wt.% RC–SG and 109.7% for the sample with equivalent RC–MMG content. Continuingly increasing the dosage of compatibilizers to 3 wt.%, the elongation at break of RC–SG- and RC–MMG-mixed samples have reached to 340.4% and 408.4%, respectively (Appendix A). Moreover, the breaking strength of RC–SG-compatibilized blends is also much higher than the blend with the equal addition of RC–MMG. For example, the breaking strength of the blend with 1 wt.% RC–SG is 46.1 MPa, while the value for RC–MMG-compatibilized blend is only 37.6 MPa. Impact toughness refers to the ability of materials to absorb plastic deformation work and fracture work under impact loading, reflecting the tiny defects and impact resistance of materials. The impact strength of the compatibilized blends is shown in Figure 5b. Both RC–SG and RC–MMG can significantly improve the impact toughness of PVDF/PLLA blend, but RC–SG always works better than RC–MMG. Such as the blends with 3 wt.% compatibilizers, the impact strength of PVDF/PLLA/RC–SG blend is 41.5 kJ/m^2^ while the value of RC–MMG-compatibilized blend is only 33.0 kJ/m^2^. This might be attributed to the co-continuous structure induced by the RC–SG compatibilizer, as shown in Figure 1. It is widely acknowledged that the co-continuous structure helps to significantly improve interfacial interactions and effectively promoting stress transfer [39]. It is also the reason for the better tensile behavior of the blend mixed with RC–SG than the RC–MMG-compatibilized blend.

### 3.3. Dynamic Mechanical Analysis

The DMA measurements were carried out for PVDF/PLLA (50/50) blends containing various types of compatibilizers. Figure 6 shows the storage modulus (E′) and tan δ as a function of the temperature. As shown in Figure 6a, the storage modulus decreases abruptly at 53 °C owing to the glass transition of PLLA for the all blends, followed by an increase of E′ originating from the cold crystallization of PLLA during heating, as evidenced by DSC curves (Appendix A). Significant difference can be observed for the storage modulus values at the temperature just above the *T*_g_ of PLLA. The RC–SG-compatibilized blends exhibit significantly higher storage modules than the RC–MMG-compatibilized blends in the temperature ranging from 60 to 100 °C, indicating the higher heat resistance of the RC–SG-compatibilized sample. This is attributed to the co-continuous microstructure of the blends. The PVDF and PLLA phases are co-continuous owing to the emulsifying effect of RC–SG. Thereby, the continuous PVDF parts still can support the blend even PLLA segments become soft above its *T*_g_. In contrast, both binary PVDF/PLLA blend and RC–MMG-compatibilized blend are the sea-island morphology with PLLA as the matrix. Therefore, both samples show lower storage modulus in the temperature region of PLLA glass transition. It should be noted that the blend without compatibilizer has a higher storage modulus than the RC–MMG-compatibilized blend. The mechanism of such difference is not clear, but it is clear that the modulus would be dependent on the domain size and domain size distribution for sea-island morphology. On the other hand, it is also seen that RC–SG induces the decreasing of the glass transition temperatures of PLLA and RC–MMG does not lead to the significant changes (Figure 6b). We used the DMA data of the samples with 1 wt.% compatibilizer to demonstrate the fact that very low compatibilizers (1 wt.%) improves the heat resistance of the blend due to the formation of co-continuous structure. In fact, the samples with a higher content of compatibilizers (3 wt.%) demonstrate similar results, as shown in Appendix A.

## 4. Discussion

It is concluded that both RC–SG and RC–MMG play the role of effective compatibilizers for PVDF/PLLA blends. They have not only the same PMMA side chains, but also similar reactivity—grafting the same amount of PLLA chains in melt-blending—thus resulting in double-graft copolymers PMMA-*g*-MMG-*g*-PLLA and PMMA-*g*-SG-*g*-PLLA, except the main chains. The PMMA and PLLA side chains of both PMMA-g-MMG-g-PLLA and PMMA-g-SG-g-PLLA can entangle with the PVDF and PLLA phase, respectively, so they can easily locate at the PVDF–PLLA interface and improve the compatibility of PVDF/PLLA blends. However, it is interesting to find that RC–SG can easily induce the stable co-continuous structure, as compared with RC–MMG. Moreover, better mechanical properties were achieved for the RC–SG-compatibilized blends with the co-continuous structure. In order to elucidate the different compatibilization mechanisms of the two compatibilizers with different main chains, more detailed morphological information was obtained by TEM, as shown in Figure 7. PVDF is observed as the black part, and PLLA corresponds to the white part, because PLLA is harder to be stained by RuO_4_ than PVDF. Clearly, PVDF forms spherical domains dispersed in the PLLA matrix for the RC–MMG-compatibilized blends. The interface is clean and sharp for the RC–MMG-compatibilized blends. In contrast, the RC–SG compatibilized blends show the elongated PVDF phase, indicating the typical co-continuous structure. In particular, such co-continuous structure means that the interface curvature is very small (or even a straight interface) for the RC–SG-compatibilized blends. We consider that the different interface curvature originates from the different main-chain structures of the two compatibilizers. PLLA chains were grafted onto the main chain of the compatibilizers during the reactive blending, so the binary grafted compatibilizers with PMMA side chains and PLLA side chains were formed as the compatibilizers for the PLLA/PVDF blends. Both compatibilizers are located at the interface through the respective entanglements (or interactions) of PLLA side chains with PLLA chains in PLLA phase and PMMA side chains with PVDF chains in PVDF phase, as shown in Scheme 2. Therefore, both RC–SG and RC–MMG play an effective role in compatibilizing the PVDF/PLLA blends. However, the different main chains of the compatibilizers lead to totally different interface curvatures of the blends and subsequent different morphologies of the final blends. For the final PMMA-*g*-SG-*g*-PLLA compatibilizers, the PS main chain was rejected by both PLLA and PVDF chains because PS shows neither interactions with PLLA nor interactions with PVDF. This is evidenced by the high immiscibility of PS/PLLA blends [40] and PS/PVDF blends [41,42]. Therefore, the PS main chain is pulled by PMMA and PLLA side chains in opposite directions, and the PS main chains form the straight conformation, as shown in Scheme 2b. Therefore, the co-continuous morphology with the straight interface can be easily obtained. In contrast, the PMMA main chains of the RC–MMG compatibilizers show highly specific interactions with PVDF phase as the side PMMA chains, so the interface with high curvatures is formed and the typical sea-island morphology is obtained. It should be noted that some very small micelles can be observed in the PLLA phase for RC–SG-compatibilized blends in Figure 7b. This originates from the unbalanced PLLA side chains and PMMA side chains of the final double-grafted molecules and such molecules were pulled into the PLLA phases. Wang et al. have synthesized similar comb-like molecules with PS main chain and PMMA side chains (*M_n_* = 2000) as compatibilizers for PVDF/PLLA blends [33]. It was found that the PS main chain of their compatibilizer collapsed to form nanomicelles at the PVDF–PLLA interface. In present work, the RC–SG has longer PMMA side chains (*M_n_* = 4000). It is considered that long side PMMA chains lead to the strong entanglements with the PVDF chains and few small micelles were observed.

## 5. Conclusions

Reactive-comb polymers with PS and PMMA main chains have been synthesized and used as the compatibilizers for the PVDF/PLLA blends. Both reactive-comb polymers exhibited drastic compatibilization effects of the blends with decreased phase size and enhanced mechanical properties. The PVDF–PLLA interface produced by RC–SG was straight due to the binary repulsion of the two side chains, helping to induce stable co-continuous morphologies. The blends with a co-continuous structure exhibited high ductility, excellent impact strength and good heat resistance.

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
