# Peer review of "Reactive Comb Polymer Compatibilized Immiscible PVDF/PLLA Blends: Effects of the Main Chain Structure of Compatibilizer"

_polymers, 2020, doi:10.3390/polym12030526_

Round 1

Reviewer 1 Report

This is a very interesting piece of work. The authors studied the effects of two double-comb compatibilizers on the morphology and mechanical properties of immiscible PVDF/PLLA blends. The main chains of the two compatibilizers are different. Both compatibilizers were found to improve the mechanical properties of the blends significantly, with RC-SG slightly more effective than RC-MMG. The results were nicely presented and discussed. The reviewer supports the publication of the manuscript.

The reviewer recommends the following minor revisions.

Scheme 1 shows that the two side chains A and B are linked to the main chain C at the same location. This can not be true. The reviewer suggests to redraw the scheme with the side chains A and B linked to the main chain at different locations.

In Figure 6a, the storage moduli of the blends were found to increase at 60-100 degree Celsius, more so for RC-MMG than for RC-SG. The authors attributed the increases to cold crystallization of PLLA. The explanation can be convincingly supported by DSC curves.

In references 20 and 21, the volume numbers of Polymer were not given.

Line 57:  chemistry is an uncountable noun, so chemistries should not be used.

Line 76 ,77 and 104: change ‘were shown’ to ‘are shown’.

Line 78: change ‘was’ to ‘is’.

Line 153: ‘poor interfacial stability’ is better than ‘bad interfacial stability’.

Author Response

Response to Reviewer #1:

This is a very interesting piece of work. The authors studied the effects of two double-comb compatibilizers on the morphology and mechanical properties of immiscible PVDF/PLLA blends. The main chains of the two compatibilizers are different. Both compatibilizers were found to improve the mechanical properties of the blends significantly, with RC-SG slightly more effective than RC-MMG. The results were nicely presented and discussed. The reviewer supports the publication of the manuscript.

The reviewer recommends the following minor revisions.

Scheme 1 shows that the two side chains A and B are linked to the main chain C at the same location. This can not be true. The reviewer suggests to redraw the scheme with the side chains A and B linked to the main chain at different locations.

Response: Thanks for your kind comments. We have carefully modified the Scheme 1. Please see Page 2, line 61.

In Figure 6a, the storage moduli of the blends were found to increase at 60-100 degree Celsius, more so for RC-MMG than for RC-SG. The authors attributed the increases to cold crystallization of PLLA. The explanation can be convincingly supported by DSC curves.

Response: Thank you very much for the suggestion. We have carried out the DSC measurements. It is observed that PLLA shows the cold crystallization at the temperature ranging from 93 oC to 120 oC. We have mentioned this in the revised version. Please see Page 9, line 217.

In references 20 and 21, the volume numbers of Polymer were not given.

Response: We have carefully checked and corrected the errors you mentioned and related content in manuscript. Thanks for your kind comments. Please see Page 13, line 355, 357.

Line 57: chemistry is an uncountable noun, so chemistries should not be used.

Response: We have checked and corrected the error.

Line 76 ,77 and 104: change ‘were shown’ to ‘are shown’.

Response: Thanks for your kind comments .We have carefully corrected the errors you mentioned in manuscript. Please see Page 2, line 76.

Line 78: change ‘was’ to ‘is’.

Response: OK. Please see Page 3, line 88.

Line 153: ‘poor interfacial stability’ is better than ‘bad interfacial stability

Response: Thank you so much for your professional advice. We have modified it in the revised version. Please see Page 5, line 163.

Reviewer 2 Report

The subject of this manuscript is interesting. Nevertheless, the paper needs an important improvement in order to be considered for publication. Please see my comments below:

English must be carefully revised. 

In the abstract, please define RC-MMG as you did with the RC-SG.

In the experimental section, it is mentioned that H-NMR and FTIR of the compatibilizers were performed but these results are presented as supplementary material, however, this information is really important and should be discussed and explained in the results not just presented as supplementary figures.

The preparation of the samples must be explained more extensively.

In the morphology results, the authors referred to thermal annealed samples. However, there is not any explanation or information about thermal annealing conditions or the reason to perform this thermal treatment. Please explain this.

In Figure 1 the caption mentioned that white bars denote 10mm, however, figure 1 a) is at different scale, please change that micrograph for others at the same scale to allow the comparison. The same occurred in figure 2 b) which its bar does not correspond to 10mm

Please modified all the micrographs of Figures 1 and 2 in order to observe the scale mark and the dimension, as it is presented in Figure 3

The tensile properties must be discussed better. Include Table S1 in the results. Compare the obtained results and explain the observations.

The discussion of impact strength results must be improved.

The dynamic mechanical analysis mentioned that significant changes are observed in the modulus and that the RC-SG compatibilized blends exhibit significantly higher storage modulus than the RC-MMG compatibilized blends, indicating the higher heat resistance of the RC-SG. However, in the non-compatibilized samples, it was obtained the same modulus than the reported for the RC-MM, in this sense, it is not that the heat resistance of RC-SG is higher. The fact is that the RC-MM is lower, please explained this.

Also, if the tensile properties showed the best results for 3% of compatibilizer content, why the DMA was performed with 1%?

Author Response

Response to Reviewer #2:

The subject of this manuscript is interesting. Nevertheless, the paper needs an important improvement in order to be considered for publication. Please see my comments below:

English must be carefully revised.

Response: Thanks for your kind comments. We have carefully checked and modified the language in this revised version.

In the abstract, please define RC-MMG as you did with the RC-SG.

Response: We have carefully modified. Please see Page 1, line 23.

In the experimental section, it is mentioned that H-NMR and FTIR of the compatibilizers were performed but these results are presented as supplementary material, however, this information is really important and should be discussed and explained in the results not just presented as supplementary figures.

Response: Thank you so much for your professional advice. We have explained this in the revised version. Please see Page 2-3, line 76-87.

The preparation of the samples must be explained more extensively.

Response: We have carefully checked and modified the preparation of the samples according to your kind suggestion. Please see Page 3, line 95.

In the morphology results, the authors referred to thermal annealed samples. However, there is not any explanation or information about thermal annealing conditions or the reason to perform this thermal treatment. Please explain this.

Response: The morphology of highly immiscible polymer blends is not stable due to the phase coalescence in the melt state. Such coalescence is originated from the high interfacial tension between the phases. Therefore, the morphological stability of polymer blends in melt indicates the effectiveness of the compatibilizers and this can usually be confirmed by the annealing experiments. We have explained it in the revised version. Please see Page 4, line 155.

In Figure 1 the caption mentioned that white bars denote 10mm, however, figure 1 a) is at different scale, please change that micrograph for others at the same scale to allow the comparison. The same occurred in figure 2 b) which its bar does not correspond to 10 mm

Please modify all the micrographs of Figures 1 and 2 in order to observe the scale mark and the dimension, as it is presented in Figure 3.

Response: We have carefully checked and corrected the figure 1 according to your kind suggestion. The co-continuous phase size of PVDF/PLLA (65/35) blend without compatibilizers is so large that cannot be clearly observed at the equal magnification as the compatibilized blends. Therefore, in order to clearly show the phase structure of the all blends, we choose the images with suitable magnifications in figure 2. We have mentioned this in the revised version. Please see page 4 line 147.

The tensile properties must be discussed better. Include Table S1 in the results. Compare the obtained results and explain the observations.

Response: Thanks for your kind comments. We have carefully modified in the revised version according to your kind suggestion. Please see Page 8, line 192.

The discussion of impact strength results must be improved.

Response: Thank you so much for your professional advice. We have improved the discussion of impact strength results in the revised version. Please see Page 8, line 201.

The dynamic mechanical analysis mentioned that significant changes are observed in the modulus and that the RC-SG compatibilized blends exhibit significantly higher storage modulus than the RC-MMG compatibilized blends, indicating the higher heat resistance of the RC-SG. However, in the non-compatibilized samples, it was obtained the same modulus than the reported for the RC-MMG, in this sense, it is not that the heat resistance of RC-SG is higher. The fact is that the RC-MMG is lower, please explained this.

Response: Thank you very much for the rational comments. The DMA results showed the highest storage modulus of RC-SG compatibilized sample at the temperature of PLLA glass transition region. This is originated from the co-continuous structure with the PVDF as the framework during the softening of PLLA components. For the samples with the sea-island morphology, the modulus would be also dependent on the morphologies, such as the domain size, size distribution, etc. We have mentioned this in the revised version. Please see line 224-229.

Also, if the tensile properties showed the best results for 3% of compatibilizer content, why the DMA was performed with 1%?

Response: Thank you very much for the rational comments. We used the DMA data of the samples with 1 wt% compatibilizer to demonstrate the fact that very low compatibilizers (1 wt%) improves the heat resistant of the blend due to the formation of co-continuous structure. In fact, the samples with higher content of compatibilizers (3wt%) show the similar results. We have added the data in the supporting information (Figure S4) in this revised version. We have also mentioned this in this revised version. Please see line 231-235.

Round 2

Reviewer 2 Report

The authors have done the corrections and the paper could be accepted.